# Potential Self-Attenuation of Arsenic by Indigenous Microorganisms in the Nakdong River

**DOI:** 10.3390/microorganisms11081910

**Published:** 2023-07-27

**Authors:** Sangmin Won, Chajeong Shin, Ho Young Kang

**Affiliations:** 1Department of Microbiology, Pusan National University, Busan 46241, Republic of Korea; 2Education/Research Group of Longevity and Marine Biotechnology for Innovative Talent, Pusan National University, Busan 46241, Republic of Korea; 3BUSAN IL Science Highschool, Busan 49317, Republic of Korea; 960283@hanmail.net

**Keywords:** heterotrophic bacteria, autotrophic bacteria, freshwater, arsenic, microbial oxidation

## Abstract

The toxic element arsenic (As) has become the major focus of global research owing to its harmful effects on human health, resulting in the establishment of several guidelines to prevent As contamination. The widespread industrial use of As has led to its accumulation in the environment, increasing the necessity to develop effective remediation technologies. Among various treatments, such as chemical, physical, and biological treatments, used to remediate As-contaminated environments, biological methods are the most economical and eco-friendly. Microbial oxidation of arsenite (As(III)) to arsenate (As(V)) is a primary detoxification strategy for As remediation as it reduces As toxicity and alters its mobility in the environment. Here, we evaluated the self-detoxification potential of microcosms isolated from Nakdong River water by investigating the autotrophic and heterotrophic oxidation of As(III) to As(V). Experimental data revealed that As(III) was oxidized to As(V) during the autotrophic and heterotrophic growth of river water microcosms. However, the rate of oxidation was significantly higher under heterotrophic conditions because of the higher cell growth and density in an organic-matter-rich environment compared to that under autotrophic conditions without the addition of external organic matter. At an As(III) concentration > 5 mM, autotrophic As(III) oxidation remained incomplete, even after an extended incubation time. This inhibition can be attributed to the toxic effect of the high contaminant concentration on bacterial growth and the acidification of the growth medium with the oxidation of As(III) to As(V). Furthermore, we isolated representative pure cultures from both heterotrophic- and autotrophic-enriched cultures. The new isolates revealed new members of As(III)-oxidizing bacteria in the diversified bacterial community. This study highlights the natural process of As attenuation within river systems, showing that microcosms in river water can detoxify As under both organic-matter-rich and -deficient conditions. Additionally, we isolated the bacterial strains HTAs10 and ATAs5 from the microcosm which can be further investigated for potential use in As remediation systems. Our findings provide insights into the microbial ecology of As(III) oxidation in river ecosystems and provide a foundation for further investigations into the application of these bacteria for bioremediation.

## 1. Introduction

The natural weathering of arsenic (As)-containing minerals and the prevalent use of As in modern industries have resulted in its excessive accumulation in the environment [1]. As contamination is a major concern worldwide owing to its detrimental effects on human health and the environment. Toxicity and remediation methods for As have been extensively studied to establish effective guidelines to alleviate the harmful effects of As contamination on humans [2]. As can cause various poisoning effects, such as nerve damage, skin damage, and increased cancer risk. Hyperkeratosis is the most common disease caused by chronic exposure to As-contaminated water [3]. 

Various techniques, including chemical, physical, and biological techniques, are available for the remediation of As contamination in the environment. Among these, biological methods are the most economical and eco-friendly for As remediation [4]. Among the two common forms of inorganic As in natural water sources, arsenite (As(III)) and arsenate (As(V)), As(III) is the most toxic and mobile form, posing a significant threat to ecosystems and drinking water supplies [3]. Microbial oxidation of As(III) to less toxic As(V) is necessary for the natural attenuation and bioremediation of As-contaminated environments [5]. Understanding the diversity and functionality of bacteria involved in this biotransformation is necessary for developing effective strategies to mitigate As pollution.

Several studies have shed light on the microbial communities and pure microbial cultures capable of As(III) oxidation, revealing the remarkable metabolic versatility of microorganisms in adapting to As-rich environments. However, most studies have focused on specific habitats, such as mine tailings [6,7,8], geothermal springs [9], and contaminated groundwater [10,11,12], with limited exploration of microbial As(III) oxidation in river water ecosystems. Rivers play vital roles in the transport and dispersion of contaminants, including As and toxic metals, highlighting their importance in understanding the microbial processes affecting As speciation and fate.

In this study, we aim to investigate the microbial oxidation of As(III) to As(V) in river water microcosms obtained from the Nakdong River and assess the As(III) oxidation potential of pure cultures isolated from the river water under both heterotrophic and autotrophic conditions. By focusing on river systems, we aimed to bridge the knowledge gap regarding microbial diversity involved in As(III) oxidation and its significance in the natural attenuation of As in aquatic environments. Additionally, identifying and characterizing bacteria capable of As(III) oxidation from river water sources can provide valuable insights into their bioremediation potential and contribute to the development of eco-friendly strategies for As removal from contaminated water. Moreover, we characterized the phylogenetic diversity of the isolated bacteria, assessed their As(III) oxidation capabilities, and explored the factors influencing their activities. 

## 2. Materials and Methods

### 2.1. Sample Collection and Preparation

River water samples were collected near the riverbank at 35°10′41.0′’ N 128°57′53.8′’ E (Figure 1). Water samples were collected in sterilized cups and placed in sterilized catcher bottles (Y&K Company, Yongin, Republic of Korea). The samples were preserved and transported to the laboratory at an ambient temperature. Then, samples were filtered through a 0.2 µm syringe filter membrane for physical and chemical analyses. The pH of the river water was measured using a professional benchtop pH meter (BP3001; Trans Instruments, TTBH Pte Ltd., Singapore) connected to a combination pH probe. Chemical elements, such as Si, S, Ca, Mg, K, and Na, were analyzed using an Inductively Coupled Plasma Optical Emission Spectrometer (ICP-OES Optima 8300; Perkin Elmer, Shelton, CT, USA). Trace elements, such as As, Se, Mo, Mn, Cu, Zn, Sb, Te, and Cr, were detected using an Inductively Coupled Plasma Triple Quadrupole-Mass Spectrometer (iCAP TQ ICP-MS; Thermo Fisher Scientific, Waltham, MA, USA).

### 2.2. As(III) Oxidation by the Indigenous Microcosm

A portion of the collected river water (20 mL), representing the river water microcosm, was inoculated into the culture medium containing 1 mM of As(III) to investigate As(III) oxidation. A minimal medium [13] containing 0.1 g/L of NH_4_Cl, 0.1 g/L of KH_2_PO_4_, 0.012 g/L of CaCl_2_·2H_2_O, and 0.05 g/L of KCl was used as the culture medium for bacterial As(III) oxidation. NaHCO_3_ (0.5 g/L) was added as the inorganic carbon source for autotrophic growth. In contrast, 1 g/L of CH_3_COONa and 1 g/L of yeast extract were added as organic carbon sources for the heterotrophic oxidation of As(III). The experiment started with an As(III) concentration of 1 mM, which was then increased to 2, 5, and 10 mM. An aliquot of the culture from the previous experiment was used as the inoculum for subsequent experiments. An abiotic control set without river water was simultaneously used in each experiment. All cultures were incubated in a shaking incubator at 25 °C and 150 rpm. Cell concentration, pH, and As(III), As(V), and total As concentrations were monitored during the experiment. All experiments were performed in triplicate, and the average values and standard deviations are presented. 

### 2.3. Bacterial Isolation and Characterization

At the end of the experiment with the indigenous microcosm containing 1 mM of As(III), a portion of the culture was serially diluted with 0.85% NaCl and spread onto agar plates containing the same ingredients as the culture medium. The agar plates were incubated at 25 °C, and colonies were found on the surface after three days of incubation. For autotrophic cultivation, colonies were selected from the 10^5^ dilution culture, whereas single colonies were selected from the 10^6^ and 10^7^ dilution cultures. The selected colonies were continuously streaked on new agar plates containing the same ingredients and As(III) at least three times to obtain pure cultures. The pure cultures of the new isolate were then transferred to a liquid medium containing 1 mM of As(III) to test their As(III) oxidation ability. Concentrations of As(III) and As(V) were determined after five days of cultivation to identify the As(III)-oxidizing bacteria among the newly isolated bacteria. 

### 2.4. As(III) Oxidation by Newly Isolated Bacteria

As(III) oxidation performance was tested using the newly isolated As(III)-oxidizing bacteria. Briefly, 1 mL of bacterial inoculum was inoculated into 100 mL of culture medium containing 1 mM of As(III). Autotrophic and heterotrophic culture media were the same as those used in the microcosm experiments. All cultures were incubated at 25 °C in a shaking incubator at 150 rpm. Abiotic controls without inoculation were run simultaneously for comparison. An aliquot of the sample was periodically extracted from each culture to monitor the pH, cell concentration, and As(III), As(V), and total As concentrations.

### 2.5. Analytical Techniques and Chemicals

During incubation, samples were collected from the culture medium and divided into three portions for subsequent analyses. The cell density of each culture was monitored by measuring the optical density of the culture medium at a wavelength of 600 nm using a digital spectrophotometer (UV–Vis Double Beam Model UVD-3200; Labomed Inc., Los Angeles, CA, USA). Each sample (1 mL), with fresh medium as a control, was added to cuvettes (Ratiolab, Szada, Hungary), and the optical density was measured against the control samples. The pH of the culture was measured by adding the culture medium (0.5 mL) to a micropH meter (LAQUAtwin-pH-22; Horiba, Kyoto, Japan). As(III) and As(V) concentrations were determined using the Dionex UltiMate 3000 high-performance liquid chromatography system (Thermo Fisher Scientific), as previously described [14,15]. Total As concentration was taken as the sum of the As(III) and As(V) concentrations. All chemicals used in this study were purchased from Sigma-Aldrich (Germany) and were of analytical grade.

### 2.6. Phylogenetic Study of Newly Isolated Bacteria

Pure colonies of newly isolated bacteria were used for DNA extraction using a DNA isolation kit for cells and tissues (Roche Diagnostics Deutschland GmbH, Mannheim, Germany). Then, DNA templates were used for *16S rRNA* gene amplification using universal primers, *27F* (5′-AGAGTTTGATCMTGGCTCAG-3′) and *1492R* (5′-TACGGYTACCTTGTTACGACTT-3′), as previously described [14]. Polymerase chain reaction products were purified and sequenced by SolGent Co., Ltd. (Daejeon, Republic of Korea). The closest relatives of the newly isolated bacteria were identified via a BLAST search of each sequence on the National Center for Biotechnology Information website. The evolutionary history of newly isolated and related strains was determined using the Mega X program [16]. A phylogenetic tree was constructed using this program. Moreover, the *16S rRNA* sequences of the newly isolated bacteria were deposited in GenBank under accession numbers OR050847 (strain HTAs10) and OR050848 (strain ATAs5).

## 3. Results and Discussions

### 3.1. River Water Characteristics

The pH of the water samples was 6.95, indicating neutral river water conditions (Table 1). Among ubiquitous elements, Ca and Na (16.5–18.9 mg/L) were in higher concentrations than other elements, such as Mg and K (3.2–3.3 mg/L). However, the concentrations of these elements were acceptable in drinking water, as per the World Health Organization (WHO) recommendation. Se, Mo, Mn, Cu, Zn, and Sb trace elements were detected in the range of 0.055–3.230 µg/L. As was detected at 0.954 g/L, which was lower than the guideline value for drinking water outlined by the United States Environmental Protection Agency and WHO. Notably, Te and Cr were not detected using the Thermo Fisher Scientific iCAP TQ ICP-MS.

### 3.2. Heterotrophic and Autotrophic As(III) Oxidation by the Indigenous Microcosm

Under heterotrophic conditions, the river water microcosm grew well and oxidized As(III) up to 10 mM (Figure 2). The concentration of As(III) remained at a steady state in the abiotic control, indicating that As(III) oxidation in these biotic systems was microbiologically catalyzed. The stability of the pH in the abiotic medium revealed that the alteration in the pH of the biotic cultures was due to microbial activities and microbially catalyzed reactions. There was no change in cell density, and the maintenance of a clear solution in the abiotic control set validated the experimental conditions used in this study.

The cell density of all cultures reached its highest value after one day of incubation and decreased slightly thereafter, except for the culture with 10 mM of As(III), where the cell density approached its highest value after three days of incubation. The highest cell density was observed in the culture with the lowest As(III) concentration (Figure 2a). The lowest cell density was observed in the culture with the highest As(III) concentration. The culture optical density of 1.0 abs was observed at the end of incubation of 1, 2, and 5 mM cultivation, whereas the culture optical density decreased up to 0.5 abs in the culture of 10 mM of As(III). This revealed that a high concentration of As might affect the growth of bacteria and partially reduce cell density. High As concentrations (10 mM) delayed the exponential phase of bacterial growth (Figure 2g). 

The pH of all culture media increased during the growth of the microcosm from 6.0 to >9.0. This may be due to the degradation of some organic matter and release of basic compounds, such as ammonia, into the heterotrophic culture medium [17,18]. Interestingly, the pH of the culture medium with 10 mM of As(III) started to decrease continuously after 6 d of incubation, correlating with the significant production of As(V) in the culture medium. In fact, the oxidation of As(III) to As(V) can result in a decrease in pH according to Equation (1) [19], as follows:AsO_2_^−^ + H_2_O + 0.5O_2_ → AsO_4_^3−^ + 2H^+^(1)

Notably, As concentrations <5 mM did not significantly affect the pH of the medium. A change in the pH of the medium resulting from the oxidation of As(III) to As(V) was observed at an As concentration of 10 mM (Figure 2g). The pH initially increased because of the breakdown of organic compounds in the heterotrophic medium but significantly decreased when more than 5 mM of As(III) was oxidized to As(V). In addition, basic compounds produced from heterotrophic media, such as ammonium, are easily volatilized when shaken for more than 10 d. This explains the difference in the pH patterns of the test with 10 mM of As and the other tests with lower As concentrations (5, 2, and 1 mM). 

The highest As(III) oxidation rate (0.71 mM/d) was observed with 5 mM of As(III). The long lag phase of As(III) oxidation was observed in the test with 1 mM of As(III) and 10 mM of As(III). At an As(III) concentration of 1 mM, the lag phase of As(III) oxidation can be explained by the acclimation stage of the microcosms required for the alteration from normal conditions to sudden contamination with As. In the test with 10 mM of As(III), the lag phase of As(III) oxidation may be due to the partial inhibition of bacterial growth by high As concentrations. This lag phase of As(III) oxidation virtually matched the long exponential phase of cell density observed in this culture. The stability of the total As concentration in all tests indicated that no As adsorption occurred in the microcosm.

The most obvious difference that can be observed easily between autotrophic and heterotrophic growth of this microcosm was the cell density of growth culture (Figure 3). The cell density during autotrophic growth was much lower than that during heterotrophic growth. At the same initial concentration of As(III) (2 mM), the cell density of heterotrophic culture was stable at around 1.0 abs, whereas that of autotrophic culture was only 0.02 abs. Autotrophic growth is always exposed to low cell density, which was also reported in a previous study on As [13,20]. Owing to low cell density, the As(III) oxidation rate can be significantly affected in autotrophic cultures.

As explained above, no organic matter was added to the autotrophic culture. Therefore, no significant increase in the pH of the autotrophic culture medium was observed during growth. In addition, a decrease in pH resulting from the oxidation of As(III) to As(V) was observed at all As concentrations. The pH dropped to pH 4.0 when a total of 2 mM of As(III) was oxidized to As(V). At higher concentrations (5 mM and 10 mM), the pH dropped to 3.0 and stabilized at a steady state. This coincides with the halting of As(III) oxidation, indicating that a pH lower than 3 may inhibit autotrophic As(III) oxidation. Without pH adjustment, only 4 mM of As(III) was oxidized to As(V), regardless of the initial concentration of As(III). The pH of the culture medium plays an important role in the microbial oxidation of As(III). 

Autotrophic As(III) oxidation occurs at a much lower rate than heterotrophic growth. At the same concentration of As (2 mM), the heterotrophic As(III) oxidation rate was more than five times higher than that of the autotrophic As(III). Because of the presence of organic compounds, the heterotrophic culture was not affected by the acidification of As(III) oxidation processes. The lowest autotrophic As(III) oxidation rate was observed in the culture with 10 mM of As, in which the microbial oxidation process was affected by the high concentrations of As and strong acidification.

### 3.3. Heterotrophic and Autotrophic As(III) Oxidation by the Newly Isolated Bacteria

Ten pure cultures were isolated from the heterotrophic enrichment, and seven pure cultures were obtained from the autotrophic enrichment. However, after testing with a culture medium containing 1 mM of As(III), only HTAs10 among the 10 isolates exhibited heterotrophic As(III) oxidation, and only ATAs5 among the seven isolates exhibited autotrophic As(III) oxidation. 

The oxidation of 1 mM of As(III) to As(V) was completed within three days with HTAs10, which was equivalent to that by the mixed microcosm (Figure 4). The cell growth pattern consistently correlated with the As(III) oxidation pattern. Approximately 80% of As(III) was oxidized on the third day of incubation. The As(III) oxidation rate of this isolate was much lower than those of previously reported heterotrophic As(III)-oxidizing bacteria, such as *Sinorhizobium* sp.KGO-5 [21], *Variovorax* sp. MM-1 [22], and *Stenotrophomonas* sp. MM-7 [23].

Interestingly, the new autotrophic As(III)-oxidizing bacterium ATAs5 could not perform as well as the mixed microcosm in which it was isolated. Only half of the added As(III) (0.5 mM) was transformed to As(III) up to 25 d (Figure 5). The pH of the medium decreased continuously during incubation. A lower cell concentration was observed compared to that of heterotrophic strains (0.01 abs). These results revealed that the key bacteria in the mixed microcosm that underwent autotrophic As(III) oxidation were not isolated. Isolation and purification of a pure culture from a mixed microcosm is impossible, and not all bacteria are culturable. Therefore, the use of a mixed microcosm is better for field applications if it shows better performance.

### 3.4. Evolutionary History of the Newly Isolated Bacteria

Phylogenetic analysis revealed that the two newly isolated bacteria belonged to two different classes (Figure 6). The phylogenetic tree clustered into two separate branches. ATAs5 belonged to Betaproteobacteria, whereas HTAs10 belonged to Actinobacteria. Both strains were classified into two bacterial phyla. ATAs5 shared a 99.86% sequence similarity (based on 100% query coverage) with the *Acidovorax radicis* strain YL-209 isolated from the Ili River Basin in China. In addition, this strain shared a high sequence similarity (based on 99% query coverage) with the *A. radicis* strain IHBB 9395 (99.64%) and *Acidovorax* sp. HP2H (99.86%). Therefore, the strain ATAs5 represents a new strain of the genus *Acidovorax*. The strain HTAs10 fell within a separate cluster of the genus *Leucobacter* of Actinobacteria. It shared a 99.64% sequence similarity (based on 100% query coverage) with *Leucobacter* sp. PH1c. In this cluster, this strain also shared a high sequence similarity (based on 99% query coverage) with *Leucobacter chromiireducens* strain Z67 (99.78%), which was isolated from the soil of the drainage ditch bank of the eighth smelter (PSW1) and *L. chromiireducens* strain L-1 (99.28%). Based on sequence similarity, the strain HTAs10 may be a new strain of the genus *Leucobacter*.

Interestingly, among the two isolated strains, the heterotrophic As(III)-oxidizing strain HTAs10 represented a new branch in the As(III)-oxidizing community. To the best of our knowledge, this is the first report on the As(III)-oxidizing ability of this genus. The strain ATAs5 belongs to Betaproteobacteria, which consists of Sb(III)-oxidizing bacteria. It is closely related to *Variovorax* sp. IDSBO-4 (99% bootstrap support) was isolated from contaminated mine sediments [7]. This strain can also grow autotrophically to oxidize Sb(III) to Sb(V). In this group, *Variovorax* sp. JL23 is a heterotrophic Sb(III)-oxidizing bacteria [24]. The close relationships between the strain ATAs5 and other Sb(III)-oxidizing bacteria, such as *Comamonas* sp. NL11 [25], *Cupriavidus* sp. NL4 [25], and *Comamonas* sp. S44 [26], were due to high bootstrap support (>50%). However, most of these bacteria are heterotrophic and Sb(III)-oxidizing in nature. Strain S44 is involved in the multifunctional metabolism of various toxic metals, such as As, Se, and Sb. 

The phylogenetic tree revealed that most As(III)-oxidizing bacteria belonged to *Alphaproteobacteria*. The two newly isolated strains in this study were new members of As(III)-oxidizing bacteria in Betaproteobacteria and Actinobacteria within the diversified microbial community. However, the diversity of bacteria involved in As(III) oxidation requires further investigation.

## 4. Conclusions

In this study, our findings demonstrated the ability of river microcosms to oxidize As(III) to As(V) under both heterotrophic and autotrophic conditions, indicating the potential self-detoxification and attenuation of As in river ecosystems. Moreover, we isolated new bacterial strains that should be further studied for future applications in As remediation systems. The internal pathways of As transformation in these microorganisms should be further investigated for appropriate applications. Our results highlight the important role of bacteria in As(III) oxidation in river water systems and emphasize their potential in mitigating As contamination. Our study can be used as a basis to develop sustainable and efficient approaches for the elimination of As pollution in natural water sources.

## Figures and Tables

**Figure 1 microorganisms-11-01910-f001:**
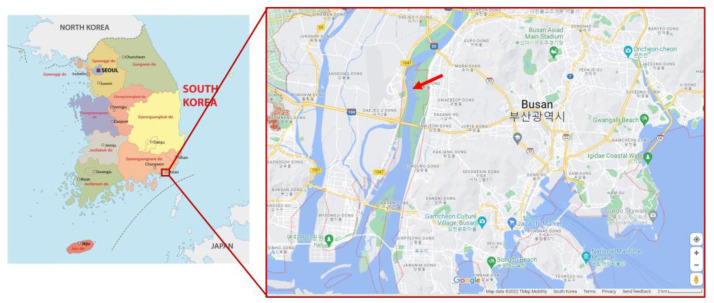
Location of the sampling site (taken from Google Maps). Red arrow indicates the sample collection site.

**Figure 2 microorganisms-11-01910-f002:**
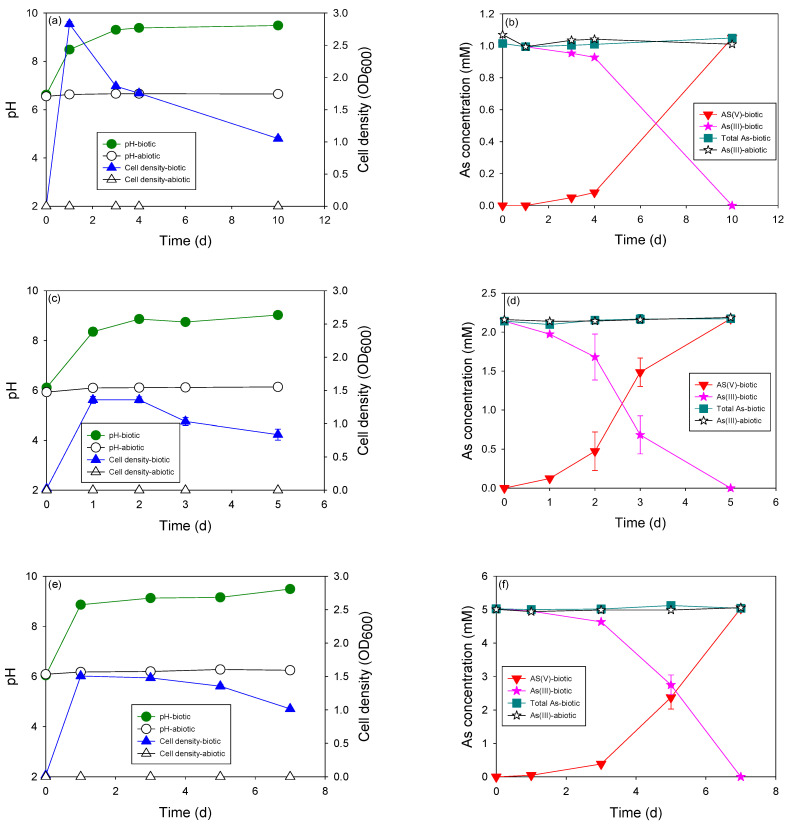
Heterotrophic arsenite (As(III)) oxidation by indigenous microcosms at various initial As(III) concentrations: (**a**,**b**) 1 mM, (**c**,**d**) 2 mM, (**e**,**f**) 5 mM, and (**g**,**h**) 10 mM.

**Figure 3 microorganisms-11-01910-f003:**
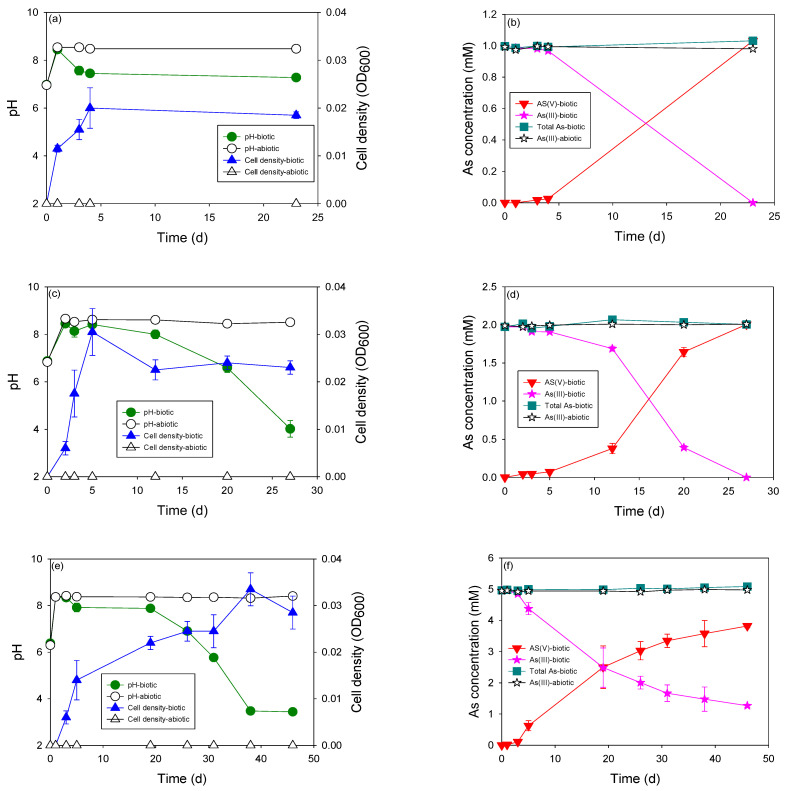
Autotrophic As(III) oxidation by indigenous microcosms at various initial As(III) concentrations: (**a**,**b**) 1 mM, (**c**,**d**) 2 mM, (**e**,**f**) 5 mM, and (**g**,**h**) 10 mM.

**Figure 4 microorganisms-11-01910-f004:**
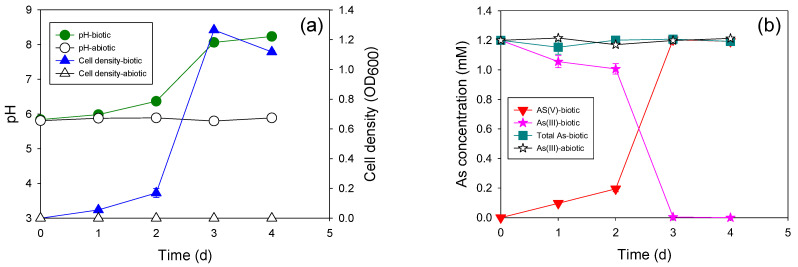
Heterotrophic As(III) oxidation with strain HTAs10 at the As(III) concentration of 1 mM. (**a**) Changes in pH and cell density; (**b**) changes in As concentration.

**Figure 5 microorganisms-11-01910-f005:**
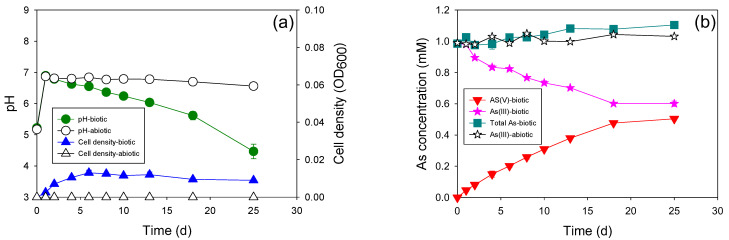
Autotrophic As(III) oxidation with strain ATAs5 at the As(III) concentration of 1 mM. (**a**) Changes in pH and cell density; (**b**) changes in As concentration.

**Figure 6 microorganisms-11-01910-f006:**
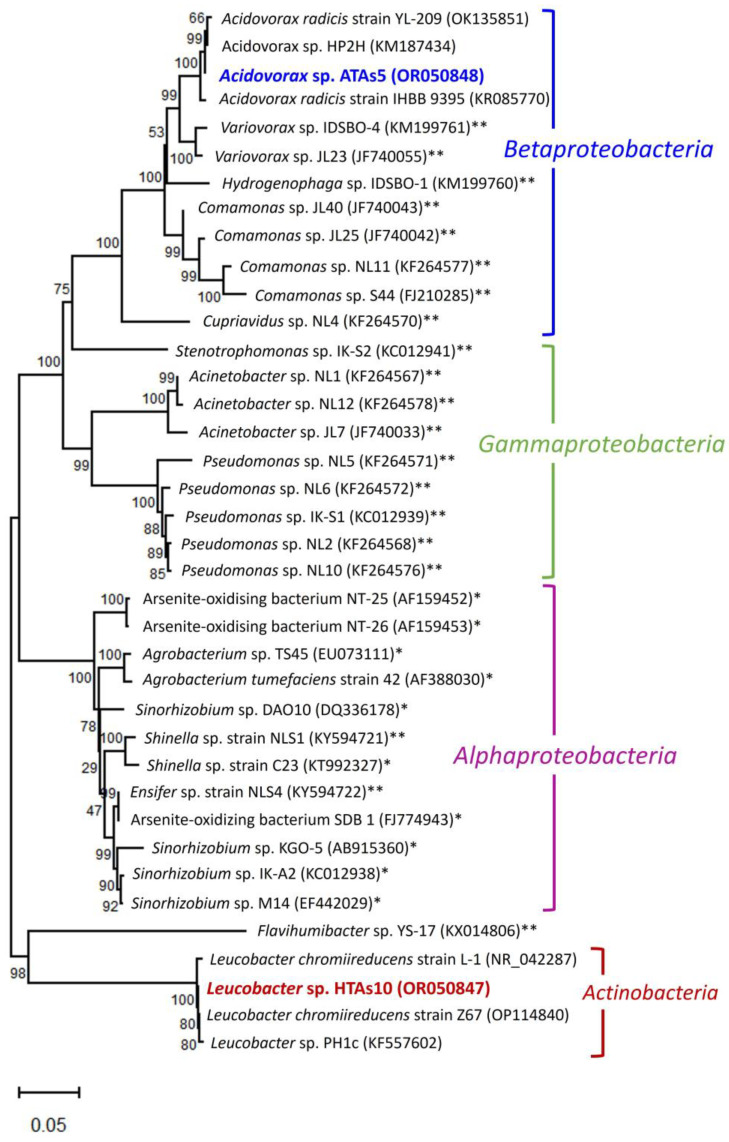
Neighbor-joining phylogenetic relationship of newly isolated bacteria with closest relatives and previously reported As(III)- and antimonite (Sb(III))-oxidising bacteria. The newly isolated strains are highlighted in red and blue colors. GenBank accession number of each strain is indicated in brackets next to the strain name. Percentages of the bootstrap test (1000 replicates) are shown next to the branches. * Previously-reported As(III)-oxidizing bacteria. ** Previously-reported Sb(III)-oxidizing bacteria.

**Table 1 microorganisms-11-01910-t001:** Physical and chemical characteristics of river water samples.

pH	Si (mg/L)	S (mg/L)	Ca (mg/L)	Mg (mg/L)	K (mg/L)	Na (mg/L)	As (µg/L)
6.95	0.921	9.528	16.518	3.333	3.297	18.932	0.954
**Se (µg/L)**	**Mo (µg/L)**	**Mn (µg/L)**	**Cu (µg/L)**	**Zn (µg/L)**	**Sb (µg/L)**	**Te (µg/L)**	**Cr (µg/L)**
0.055	2.226	3.230	1.752	0.815	0.424	ND ^1^	ND

^1^ ND: non-detected.

## Data Availability

Data is unavailable due to privacy.

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
