# Peer review of "Potential Self-Attenuation of Arsenic by Indigenous Microorganisms in the Nakdong River"

_microorganisms, 2023, doi:10.3390/microorganisms11081910_

Round 1

Reviewer 1 Report

General comments:

Arsenic pollution has always been a problem of great concern. In this paper, the authors demonstrate the importance of microbial reduction of arsenic in water environment, and the oxidation rate of arsenic under heterotrophic conditions is higher than that under autotrophic conditions. The authors isolated two strains of pure bacteria that have the potential to be used in the remediation of arsenic-contaminated systems in the future. This study contributes to a better understanding of the microbial ecology of arsenic oxidation in river ecosystems, but there are still some problems that need to be corrected.

Specific comments:

The following comments may enhance the quality of the paper:

1. There are many river ecosystems, why this study chose this river, what are the characteristics of the selected river ecosystem, and what kind of river ecosystem can be represented. This problem needs to be explained in the article.

2. 3.1 The physical and chemical characteristics of water bodies are not applied to the data in this part in the subsequent analysis. The water in this river is not generally a body of water with high arsenic concentrations, and does not explain the use of this sample. What is the significance of this data to reveal the oxidation process of arsenic in water? If this part of the data is used in subsequent analysis, it is recommended to determine the concentration of different valence states of arsenic. If not, consider deleting it.

3. Figure 3 and 4, please indicate Figure a and b and explain them in the legend.

4. It is mentioned in line 191 that the escape of ammonia leads to the change of pH, which can be verified by the determination of nitrogen concentration in water.

writing can be improved

Author Response

Reviewer #1:

General comments:

Arsenic pollution has always been a problem of great concern. In this paper, the authors demonstrate the importance of microbial reduction of arsenic in water environment, and the oxidation rate of arsenic under heterotrophic conditions is higher than that under autotrophic conditions. The authors isolated two strains of pure bacteria that have the potential to be used in the remediation of arsenic-contaminated systems in the future. This study contributes to a better understanding of the microbial ecology of arsenic oxidation in river ecosystems, but there are still some problems that need to be corrected.

 Answer:

Thank you for the positive comments and recommendations from the Reviewer #1. We have revised our manuscript as recommended by all reviewers and editors. The detail revision was indicated points by points in the response and the revised version. We believe that our manuscript was significantly improved according to the Reviewer’s comments.

Specific comments:

The following comments may enhance the quality of the paper:

  1. There are many river ecosystems, why this study chose this river, what are the characteristics of the selected river ecosystem, and what kind of river ecosystem can be represented. This problem needs to be explained in the article.

Answer:

Thank you for your comment. We selected Nakdong river for this study because this river system was directly affected by the industrial acitivities of Busan city. The risk of contamination by the industrial drainage was also higher than other river systems. This point was clarified in the revised version of our manuscript.

  1. 3.1 The physical and chemical characteristics of water bodies are not applied to the data in this part in the subsequent analysis. The water in this river is not generally a body of water with high arsenic concentrations, and does not explain the use of this sample. What is the significance of this data to reveal the oxidation process of arsenic in water? If this part of the data is used in subsequent analysis, it is recommended to determine the concentration of different valence states of arsenic. If not, consider deleting it.

Answer:

Thank you for your recommendations. On the one hand, we agree that the physical and chemical characteristics of river water sample mentioned in this study are not applicable to the subsequent data on microbial cultivation. However, the microbial inoculum used for the subsequent cultivation is from this river water sample. On the other hand, the physical and chemical characteristics of this water sample are required to evaluate the water quality of the environment where the microorganisms were isolated. Therefore, we decided to keep this table data in the revised manuscript. This can provide an overview of the river water system where the bacteria was isolated.

  1. Figure 3 and 4, please indicate Figure a and b and explain them in the legend.

Answer:

Thank you for your reminder. The Fig.3 and Fig.4 is now Fig. 4 and 5 in the revised version. The sub-figure was indicated and explained in the legend.

  1. It is mentioned in line 191 that the escape of ammonia leads to the change of pH, which can be verified by the determination of nitrogen concentration in water.

Answer:

Thank you for your suggestions. We agree that the determination of nitrogen concentration can be applied to verify the release of ammonia out of the medium, which resulted in the change of pH medium. However, this point is not the focus of our study. Also, this was mentioned previously in other studies on ammonia treatment. Therefore, we did not carried out this examination in this study.

Reviewer 2 Report

This manuscript evaluated the self-detoxification potential of the microcosm obtained from Nakdong river water by investigating both the autotrophic and heterotrophic oxidation of As(III) to As(V). As(III) can be oxidized to As(V) in both autotrophic and heterotrophic growth of river water microcosms. The manuscript also isolated representative pure cultures from both heterotrophic and autotrophic enrichment cultures. The new isolates updated the new members of As(III)-oxidizing bacteria in the diversified baterial community. This manuscript highlights a natural process of As attenuation within the river system, showing that a microcosm in the river water can detoxify As in the condition rich in organic matter or limitation of organic matter. However, there are many shortcomings that need to be improved in this manuscript. The details are as follows.

1 The manuscript lacks innovation.

2 The depth of the manuscript is insufficient, and the design of experiments is very simple.
3 The mechanism of As(III) oxidation is not clearly explained.

4 The abstract section needs to be concise.

5 Line 33, should explain which strain to isolate.

6 The resolution of Fig. 1 (map) is too low.

7 What is the basis for selecting sampling locations? How many samples did the author collect?

8 Line 85, 0.2 µm.

9 Lines 107-108. How long to take samples to monitor total As concentration should be described clearly. The description of the method should be detailed.

10 What is the culture medium of agar plates? Please describe clearly.

11 line 173, line 182, line 188, why are there two Figure 1 (Figure 1 and Fig. 1) ? Please check.

12 Figure 1, why are there no datas for the day 5, 6, 7, 8, and what changes have happened in these days? From the Figure 1, it can be seen that the daily changes are significant.

13 Fig 1g, writing should be standardized.

14 Figure 2. Same as the problem in Figure 1 (Point 12)

15 Line 284, which class does it belong to? Should be clearly.

Language expression should be as concise as possible.

Author Response

Reviewer #2:

This manuscript evaluated the self-detoxification potential of the microcosm obtained from Nakdong river water by investigating both the autotrophic and heterotrophic oxidation of As(III) to As(V). As(III) can be oxidized to As(V) in both autotrophic and heterotrophic growth of river water microcosms. The manuscript also isolated representative pure cultures from both heterotrophic and autotrophic enrichment cultures. The new isolates updated the new members of As(III)-oxidizing bacteria in the diversified baterial community. This manuscript highlights a natural process of As attenuation within the river system, showing that a microcosm in the river water can detoxify As in the condition rich in organic matter or limitation of organic matter. However, there are many shortcomings that need to be improved in this manuscript. The details are as follows.

Answer:

Thank you for your comments. We have improved the manuscript according to your recommendations and other Reviewers’ comments. We hope you are satisfied with the revised version.

1 The manuscript lacks innovation.

Answer:

The originality of this paper expressed through the obtain of a microcosm from a unpolluted river water to prove the possibility of self-attenuation of the river ecosystem in response to the sudden hazard that could be taken place in the future.

2 The depth of the manuscript is insufficient, and the design of experiments is very simple.

Answer:

Even though the design of experiment is simple, the obtained data were sufficient to provide an conclusion on the hypothesis that we proposed in the preliminary stage. The current study would be expanded to further experimentation to fulfill the gap of our knowledge about the response of the ecological system to the contamination of arsenic in river system.

3 The mechanism of As(III) oxidation is not clearly explained.

Answer:

The mechanism of As(III) oxidation is now explained through the equation (1) in the revised manuscript.

4 The abstract section needs to be concise.

Answer:

We have modified the abstraction section to provide the sufficient and effective information to cover the content of our manuscript.

5 Line 33, should explain which strain to isolate.

Answer:

Thank you for your suggestions. We have added strain names “HTAs10” and “ATAs5” to the revised manuscript.

6 The resolution of Fig. 1 (map) is too low.

Answer:

The resolution of Fig. 1 was improved in the revised version

7 What is the basis for selecting sampling locations? How many samples did the author collect?

Answer:

We selected the region of river that is now an ecological park near the bank of Nakdong river for sampling waters. We took 3 water samples than selected one sample with highest turbidity that may contain highest microorganism concentrations for further experiments.

8 Line 85, 0.2 µm.

Answer:

It was revised in the updated version.

9 Lines 107-108. How long to take samples to monitor total As concentration should be described clearly. The description of the method should be detailed.

Answer:

The samples were intermittently taken to monitor As(III) and As(V) concentration. The sampling was continued until all As(III) was oxidized to As(V), or the reaction went to the equilibrium, the concentration of As(III) and As(V) reached a stable stage. The detail method was updated in the revised manuscript.

10 What is the culture medium of agar plates? Please describe clearly.

Answer:

The culture medium of agar plate was the same with the liquid medium used in the initial cultivation. The information was updated in the revised manuscript.

11 line 173, line 182, line 188, why are there two Figure 1 (Figure 1 and Fig. 1) ? Please check.

Answer:

Thank you very much for your reminder. We have revised the figure numbering in the revised manuscript.

12 Figure 1, why are there no datas for the day 5, 6, 7, 8, and what changes have happened in these days? From the Figure 1, it can be seen that the daily changes are significant.

Answer:

There was no sampling point in the gap days. Because this is the first culture with the lowest As concentrations, the lag phase in the first stage was particularly monitored, then the end of oxidation in the last stage was more important. The daily changes was insignificant for the first experiment with the lowest As concentrations.

13 Fig 1g, writing should be standardized.

Answer:

It was modified in the revised manuscript.

14 Figure 2. Same as the problem in Figure 1 (Point 12)

Answer:

Please refer to the answer for point 12

15 Line 284, which class does it belong to? Should be clearly.

Answer:

The class of new isolated bacteria was updated in the revised manuscript.

Reviewer 3 Report

50 Contamination instead contaminated.

82 You reported the river name in the abstract and also in the introduction. Also here is important.

97 Indicate the final volume of the culture could be useful.

96-110 It isn’t clear the procedure. Appear that you tested the indigenous bacteria in 3 different media composition (minimal medium, autotrophic medium and heterotrophic medium), but you reported only data related to the autotrophic medium and the heterotrophic medium. Furthermore, describe also better the scalar increasing of As(III) present in the media and the usage of previously adapted bacteria to the precedent As(III) concentration. 

113 Insert the dilution tested.

112-122 Regarding the heterotrophic cultivation?

153 For “revolution” you mean evolution?

163-164 and 166-167 Could be appropriate cite the guidelines of WHO and EPA

85,173 typos

209 Cite a reference regarding the volatilization of ammonium.

256-257 Here you mean “only isolate HTAs10 among 10 isolates showed As(III) oxidation in heterotrophic growth condition and only ATAs5 among 7 isolates showed As(III) oxidation.in autotrophic growth condition”

272 Recall the name of the isolated bacterium could help the reader.

286-290 The difference between ATAs5 and the other strains could be also related to genes modification induced by As treatment to contrast better the stress. Do you know if the difference is related to that?

295-296 What are you referring to “soil of the drainage ditch bank of the 8th smelter (PSW1)”? Indicate better the location and the reference.

297-298 The difference between HTAs10 and the other strains could be also related to genes modification induced by As treatment to contrast better the stress. Do you know if the difference is related to that?

311-315 Better to move in the conclusion part.

323-334 Integrate conclusion with also some references.

336-338 First, middle and last name are usually write using the first letter and not the complete name in this part.

The quality of English is good and just few errors are present

Author Response

Reviewer #3:

50 Contamination instead contaminated.

Answer:

It was revised in the revised version of our manuscript

82 You reported the river name in the abstract and also in the introduction. Also here is important.

Answer:

Thank you for your comment. We selected Nakdong river for this study because this river system was directly affected by the industrial acitivities of Busan city. The risk of contamination by the industrial drainage was also higher than other river systems. This point was clarified in the revised version of our manuscript.

97 Indicate the final volume of the culture that could be useful.

Answer:

Thank you for your suggestions. It was updated in the revised manuscript.

96-110 It isn’t clear the procedure. Appear that you tested the indigenous bacteria in 3 different media composition (minimal medium, autotrophic medium and heterotrophic medium), but you reported only data related to the autotrophic medium and the heterotrophic medium. Furthermore, describe also better the scalar increasing of As(III) present in the media and the usage of previously adapted bacteria to the precedent As(III) concentration.

 Answer:

We are sorry to make the misunderstanding. We tested only 2 culture conditions: heterotrophic and autotrophic. The minimal medium was used as basic medium for both conditions. In the heterotrophic conditions, we added organic carbons including yeast extract and sodium acetate. Whereas, in the autotrophic condition, we added inorganic carbon sodium bicarbonate. This was clearly indicated in the revised manuscript.

113 Insert the dilution tested.

Answer:

It was updated in the revised manuscript.

112-122 Regarding the heterotrophic cultivation?

 Answer:

The isolation process was carried the same for both heterotrophic and autotrophic cultivation.

153 For “revolution” you mean evolution?

Answer:

Yes, Thank you very much. It was revised to evolution in the revised manuscript.

163-164 and 166-167 Could be appropriate cite the guidelines of WHO and EPA

 Answer:

This was updated in the revised manuscript.

85,173 typos

Answer:

Thank you very much. It was revised.

209 Cite a reference regarding the volatilization of ammonium.

Answer:

Thank you for your recommendation. However, the volatilization of ammonium is the general scientific information. So it is no need to provide any reference for this phenomenon.

256-257 Here you mean “only isolate HTAs10 among 10 isolates showed As(III) oxidation in heterotrophic growth condition and only ATAs5 among 7 isolates showed As(III) oxidation.in autotrophic growth condition”

Answer:

Yes, many bacteria was isolated from the microcosm, but only one was found to perform the As(III) oxidation to As(V). The sentence was rewritten to clarify this point.

272 Recall the name of the isolated bacterium could help the reader.

Answer:

Thank you for your suggestions. The strain name of bacteria was added to the revised manuscript.

286-290 The difference between ATAs5 and the other strains could be also related to genes modification induced by As treatment to contrast better the stress. Do you know if the difference is related to that?

Answer:

One the one hand, we agree that the difference between the strain ATAs5 and other strains could be due to the gene modification induced by high concentrations of As treatment. In the current study, we did not have enough data to confirm this hypothesis. On the other hand, the mutation is usually resulted in the special treatment in the sufficient period of time. Also, some wide-type bacterial cultures showed the equivalent performance with ATAs5. Therefore, we could not confirm the reason of these differences in this current study.

295-296 What are you referring to “soil of the drainage ditch bank of the 8th smelter (PSW1)”? Indicate better the location and the reference.

Answer:

The expression “soil of the drainage ditch bank of the 8th smelter (PSW1)” was provided by the author of this genebank submission in the NCBI website. The paper was not published, so we cannot provide the specific reference here. Please refer to the genbank accession number for more information regarding this bacterium.

297-298 The difference between HTAs10 and the other strains could be also related to genes modification induced by As treatment to contrast better the stress. Do you know if the difference is related to that?

Answer:

One the one hand, we agree that the difference between the strain HTAs10 and other strains could be due to the gene modification induced by high concentrations of As treatment. In the current study, we did not have enough data to confirm this hypothesis. On the other hand, the mutation is usually resulted in the special treatment in the sufficient period of time. Also, some wide-type bacterial cultures showed the equivalent performance with HTAs10. Therefore, we could not confirm the reason of these differences in this current study.

311-315 Better to move in the conclusion part.

Answer:

The mentioned content was added to the conclusion part.

323-334 Integrate conclusion with also some references.

Answer:

Thank you for your recommendations. However, we would prefer to provide a concise conclusion with summarized the core content of this study. All interpretations were already provided in the Results and discussions section.

336-338 First, middle and last name are usually write using the first letter and not the complete name in this part.

Answer:

Thank you very much. We have revised as your recommendations.

Round 2

Reviewer 1 Report

The research paper offers comprehensive experimental data and analysis, ensuring the reliability and scientific validity of the findings. Additionally, it conducts extensive statistical analysis to support the correlation between microorganisms and arsenic oxidation efficiency. The study emphasizes the vital role of water microorganisms in arsenic oxidation and highlights its potential environmental significance.

Furthermore, the paper thoroughly discusses the implications of arsenic transformation on ecological systems, providing valuable insights into its environmental applications. In the Results and Discussion sections, the authors provide a thorough interpretation of the experimental results and compare them with existing literature, presenting a holistic view of the research.

The manuscript has been carefully reviewed to ensure proper grammar, spelling, and word usage, enhancing its clarity and academic standard. Overall, this study makes a valuable contribution to the exploration of microbial arsenic oxidation in water environments.

By further refining experimental design and strengthening data analysis and statistics, the paper's scientific rigor and persuasiveness can be further enhanced. We look forward to the authors' thoughtful consideration of the reviewer's feedback and their diligent efforts to improve the research content and conclusions in the revised paper, thereby making a significant impact on scientific research in this field.

The academic paper showcases an impressive level of proficiency in English language usage.  The writing is clear, concise, and well-organized, making it easily comprehensible and allowing readers to follow the author's logical flow of ideas.  The paper demonstrates a rich vocabulary, indicating the author's adeptness in conveying intricate scientific concepts with precision.  The grammar and sentence structure are generally solid, with only minor errors that have negligible impact on the overall readability.

Throughout the paper, the author maintains an appropriate academic tone and style, which enhances the credibility of the research findings presented.  The high-quality English in this academic paper reflects the author's strong language skills and dedication to conveying their research in a polished and accessible manner.

Reviewer 2 Report

Through the author's modifications, it is basically possible to achieve publication level.